# Reporting of harms data in RCTs: a systematic review of empirical assessments against the CONSORT harms extension

Alex Hodkinson, Jamie J Kirkham, Catrin Tudur-Smith, Carrol Gamble

CT-S and CG are joint senior authors

Department of Biostatistics, MRC North West Hub for Trials Methodology Research, University of Liverpool, Liverpool, UK

**Correspondence to**
Alex Hodkinson;
ahoddy@liverpool.ac.uk

## ABSTRACT

**Objective:** To determine the standard of reporting of harms-related data, in randomised controlled trials (RCTs) according to the Consolidated Standards of Reporting Trials (CONSORT) statement extension for harms.

**Design:** Systematic review.

**Data sources:** The Cochrane library, Ovid MEDLINE, Scopus and ISI Web of Knowledge were searched for relevant literature.

**Eligibility criteria for selecting studies:** We included publications of studies that used the CONSORT harms extension to assess the reporting of harms in RCTs.

**Results:** We identified 7 studies which included between 10 and 205 RCTs. The clinical areas of the 7 studies were: hypertension (1), urology (1), epilepsy (1), complimentary medicine (2) and two not restricted to a clinical topic. Quality of the 7 studies was assessed by a risk of bias tool and was found to be variable. Adherence to the CONSORT harms criteria reported in the 7 studies was inadequate and variable across the items in the checklist. Adverse events are poorly defined, with 6 studies failing to exceed 50% adherence to the items in the checklist.

**Conclusions:** Readers of RCT publications need to be able to balance the trade-offs between benefits and harms of interventions. This systematic review suggests that this is compromised due to poor reporting of harms which is evident across a range of clinical areas. Improvements in quality could be achieved by wider adoption of the CONSORT harms criteria by journals reporting RCTs.

## INTRODUCTION

Every healthcare intervention is associated with a risk of harmful or adverse events, that must be balanced against the potential favourable outcomes.[1]

The Consolidated Standards of Reporting Trials (CONSORT) statement aims to improve the quality of published reports of randomised controlled trials (RCTs) and has

---

### ARTICLE SUMMARY

**Strengths and limitations of this study**

- This is the first study to systematically review empirical studies assessing the quality of reporting according to the CONSORT-harms guideline.
- The review was strengthened by its assessment of quality of the included studies across four key domains.
- This study should be regarded as a reflection of reporting standards in general rather than an assessment of adherence to the CONSORT-harms extension.
- Some included studies contained trials reported prior to the publication of the CONSORT harms guideline; we did not extract these results.
- We have not assessed changes in reporting over time.

---

been widely endorsed by healthcare journals, leading to improvements in quality when used by manuscript authors and peer reviewers.[2–4] However, some reports suggest that assessment and reporting of harms in clinical trials may be suboptimal.[5–7]

The standard CONSORT statement[8] is primarily aimed at reporting the intended, usually beneficial effects of intervention(s) with only one item (item 19) devoted to unintended adverse events (harms) in the original 2001 checklist. Owing to accumulating evidence that reporting on harms-related data in RCTs was of poor quality with an imbalanced ratio of benefit–harms reporting, a CONSORT statement extension was developed in 2004 to improve harms reporting (CONSORT-harms) and to help address perceived shortcomings in measurement, analysis and reporting of harms data.[9] The subsequent update of the standard CONSORT statement, published in 2010,[10] now specifically refers to the additional CONSORT-harms extension but it is still unclear whether authors and journals routinely adopt the use of this extension. The aim

of this paper is to systematically review the evidence from previously conducted empirical studies that have assessed the adequacy of harms reporting in RCTs using the CONSORT-harms extension as a benchmark.

## METHODS

A protocol for the systematic review was developed by AH, CTS, CG and JJK.

### Study inclusion criteria

We included published and unpublished research that evaluated the quality of harms reporting in RCTs against the CONSORT-harms recommendations.[9] No restriction was placed on the clinical area or type of intervention studied. Excluded studies were those that assessed harms reporting using assessment criteria other than CONSORT-harms and studies that assessed harms reporting using study designs for which the CONSORT guideline was not intended (eg, observational studies).

### Identification of studies

AH, CTS and CG developed the search strategy with support from an information specialist which was then implemented by AH in the following databases: the Cochrane Methodology register, Database of Abstracts of Reviews of Effects (DARE), Ovid MEDLINE, Scopus and ISI Web of Knowledge. Conference abstracts were searched for in the Web of Knowledge Conference Proceedings Citation Indexes (CPCI-S or CPCI-SSH) and the Zetoc database.[11] An unpublished Masters dissertation involving one of the authors (JJK) was also obtained. Date filters were not used during the search criteria; our interest lies only within reviews published after 2004, with the cut-off date June 2012.

Titles and abstracts of reports identified by the search were screened by AH and full articles obtained for all potentially eligible studies. Each full article was assessed independently by two reviewers (AH and CTS) to determine eligibility.

### Quality assessment

Two reviewers (AH and JJK) independently assessed the methodological quality of each study using the Cochrane Risk of Bias (RoB) tool[12] as a guideline to cover the following aspects. Criteria were graded as low risk, high risk or unclear as indicated.

1. Were the trials included in the study a representative sample, for example, unselected journals, and reasonable time scale?
   Low risk of bias: studies included trials from a primary search of all the available literature.
   High risk of bias: studies were highly selective of the trials included, for example, high-impact journals or specialised-journals only.
   Unclear risk of bias: not stated how studies were selected.

2. During the data extraction of CONSORT-harms criteria, were reviewers blinded to study authors, institution, journal name and sponsors?
   Low risk of bias: reviewers were blinded.
   High risk of bias: reviewers were not blinded.
   Unclear risk of bias: not stated.

3. Is there evidence of selective outcome reporting in the study (ie, were all CONSORT-harms recommendations considered and if not were suitable reasons provided)?
   Low risk of bias: studies that considered all CONSORT-harms criteria or reasons for excluding specific criteria were transparent and justified.
   High risk of bias: studies did not consider all CONSORT-harms criteria.
   Unclear risk of bias: unclear whether all CONSORT-harms criteria were considered.

4. Did more than one reviewer assess the CONSORT-harms criteria for each primary RCT, with a description of how agreement was achieved?
   Low risk of bias: data extraction was completed independently by two people or reasonable attempts were made to maximise data extraction reliability.
   High risk of bias: data extraction not completed independently by two people.
   Unclear risk of bias: not stated.

### Data collection and extraction

Two reviewers (AH and JJK) independently extracted the data and any discrepancies were resolved through a consensus discussion with a third reviewer (CTS). Data extraction included

▶ Study characteristics: inclusion criteria including clinical area, types of interventions, databases or journals searched within the study and any search date restrictions.
▶ Sample size (defined by the number of RCT reports assessed for reporting quality).
▶ Reporting quality: inclusion of any of the 10 recommendations from the 2004 CONSORT-harms checklist (table 1 and supplemental data: see CONSORT plots).

Lead authors were contacted through email with any queries relating to the quality assessment or data extraction.

### Data analysis and presentation

For each study, the percentage of included RCTs that satisfied each CONSORT-harms recommendation is presented with 95% CIs. Some studies had presented data for individual items described within each of the 10 criteria rather than overall data. These are presented as such in tables with footnotes to provide further explanation. Forest plots were used to graphically depict the levels of adherence to the CONSORT harms recommendations so that readers can easily discern the extent of compliance and heterogeneity between studies with the $I^2$ statistic (included as supplementary material online). We refrained from statistically combining results from the different studies due to the differences in their study characteristics. In accordance

**Table 1** The 10 CONSORT-harms recommendations[9]

| Recommendation | Description |
|---|---|
| 1 | If the study collected data on harms and benefits, the title and abstract should so state |
| 2 | If the trial addresses harms as well as benefits, the introduction should so state |
| 3 | List addressed adverse events with definitions for each (with attention, when relevant, to grading, expected vs unexpected events, reference to standardised and validated definitions, and description of new definitions) |
| 4 | Clarify how harms-related information was collected (mode of data collection, timing, attribution methods, intensity of ascertainment, and harms-related monitoring and stopping rules, if pertinent) |
| 5 | Describe plans for presenting and analysing information on harms (including coding, handling of recurrent events, specification of timing issues, handling of continuous measures and any statistical analyses) |
| 6 | Describe for each arm the participant withdrawals that are due to harms and the experience with the allocated treatment |
| 7 | Provide the denominators for analyses on harms. |
| 8 | Present the absolute risk of each adverse event (specifying type, grade, and seriousness per arm), and present appropriate metrics for recurrent events, continuous variables and scale variables, whenever pertinent. |
| 9 | Describe any subgroup analyses and exploratory analyses for harms |
| 10 | Provide a balanced discussion of benefits and harms with emphasis on study limitations, generalisability and other sources of information on harms |

with the Cochrane Handbook, $I^2$ statistics were interpreted as (0–40%, might not be important; 30–60%, may represent moderate heterogeneity; 50–90% may represent substantial heterogeneity; 75–100%, considerable heterogeneity).[13]

## RESULTS
The search strategy identified 5083 potentially eligible study cohorts from which seven studies assessing the quality of reporting across almost 800 RCTs were included (figure 1).

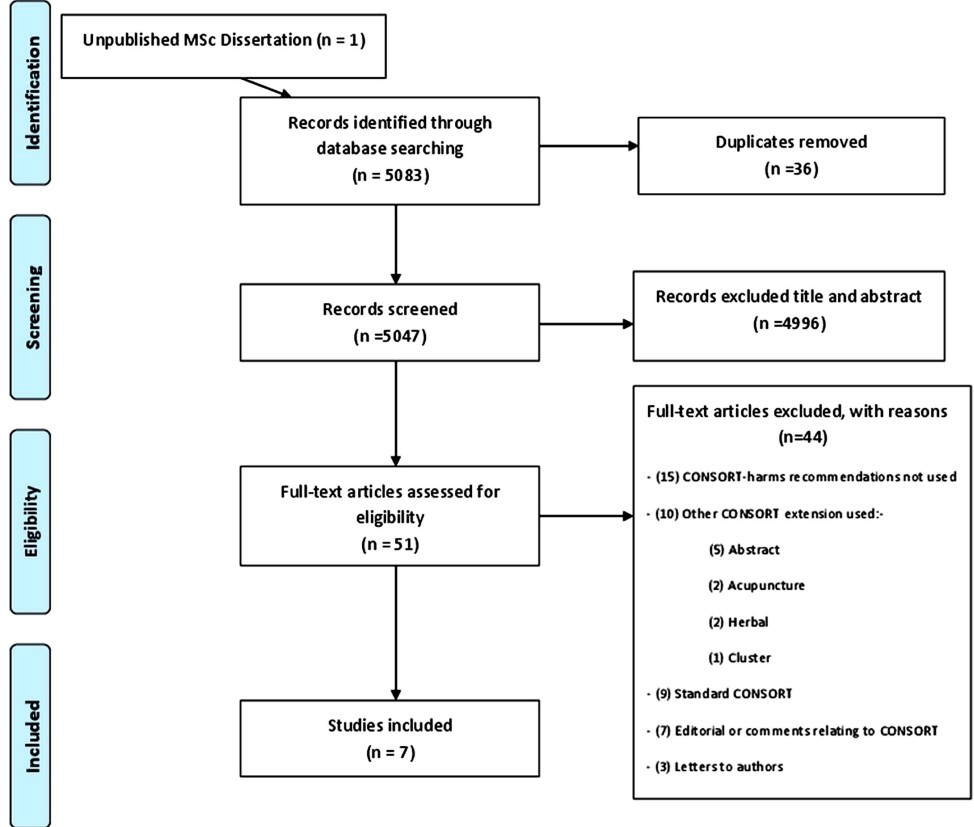

**Figure 1** Flow diagram of study identification and selection.

**Table 2**  Characteristics of included reviews

| Review characteristic | Bagul and Kirkham[14] | Breau et al[15] | Turner et al[16] | Shukralla et al[17] | Capili et al[18] | Pitrou et al[20] | Haidich et al[21] |
|---|---|---|---|---|---|---|---|
| Clinical area | Hypertension | Urology | Acupuncture therapies and other complementary alternative medicines (CAM) | Epilepsy | Acupuncture | Mixed | Mixed |
| Type of intervention(s) | Drug interventions for hypertension | Unclear | Acupuncture, massage therapies and herbal medicines. | Drug interventions for epilepsy | Acupuncture (excluding studies that evaluated acupressure, laser acupuncture, and auricular acupuncture) | Drug interventions | Drug interventions |
| Journals/databases searched | Cochrane Cent Regis | *J Urol*; Urology; Eur Urol; BJU Int | Cochrane Collaboration's CAM Field Specialised Regist Trials | MEDLINE; Cochrane Libr; Epilepsy Group Trial Registry | MEDLINE; Allied Complement Med; Cumulative Index Nurs Allied Health Lit; Evid Based Med Rev | MEDLINE via PubMed (NEJM, Lancet, JAMA;BMJ; Ann Intern Med; PLoS Med) | MEDLINE (Ann Intern Med, BMJ, JAMA, Lancet, NEJM) |
| Date restrictions | January 2005 to September 2010 | 1996 and 2004 only | 2009 | January 1999 to December 2008 | 2005 to 2008 | 1 Jan 2006 to 1 Jan 2007 | 2003 and 2006 only |
| Number of RCTs included in the review | 41 | 152 | 205 | 152 | 10 | 133 | 102 |
| Inclusion criteria | Randomised controlled hypertension trials comparing two parallel arms reported in standard CONSORT endorsing journals | RCTs of therapeutic interventions published in the three selected journals for the 2 years. | All Cochrane Complementary Medicine RCTs pertaining to 15 CAM intervention categories. | RCTs comparing AEDs (antiepileptic drugs); RCT patient population with epilepsy; RCTs published in English. | Studies published in the English language, acupuncture for pain reduction, a method for evaluating level of pain, and randomised allocation to treatment group. | Articles were included if the study was identified as an RCT with two parallel arms (in selected journals). | Published RCTs assessing drugs in the selected journals for the 2 years. |

Five studies[14–18] (with the study[19] recently published) contained trials focusing on specific clinical areas with two[20 21] covering multiple clinical areas (table 2). Four studies[14 17 20 21] included trials using drug interventions, one comparing acupuncture[18] and another alternative complementary medicines,[16] the interventions were unclear in one study.[15] MEDLINE was used by four of the studies[17 18 20 21] to identify the relevant literature; three[14 16 17] used the Cochrane database of RCTs and three[15–17] searched specialised-journal databases. The date restrictions used in the search strategy of each study ranged from a 1 year period up to a 9 years span. The

studies were published after 2008, 4 years after the release of the harms extension with three[15 17 21] of them including trials that had been published before the publication of CONSORT-harms. Five studies[14–17 20] excluded trials published in a non-English language.

## Risk of bias

Lead authors were contacted by email with any queries relating to the quality of their study, or CONSORT criteria; however, two authors failed to respond.[15 18] The risk of bias for the seven included studies, assessed across four domains, is summarised in table 3. Six studies[14 15 17 18 20 21] were classified as high risk for bias for at least one domain with two of these studies[14 20] classified as high risk for three domains. Four studies[13 14 20 21] did not include trials of a representative sample targeting specific journals rather than a database search. Blinding of assessors was only implemented in two studies[15 16] with one unclear.[18] Most studies used all the CONSORT-harms criteria with the exception of the subgroup analysis item; one study[16] however, discarded the use of recommendation eight, since it was captured elsewhere within the data extraction, and recommendation 10, which was considered too vague to assess with any objectivity. Reporting of the assessment within three[15 18 20] of the seven identified studies was unclear and authors were contacted. The authors did not respond for two studies[15 17] and in another study[20] a response was received but some details remained unclear. Six studies[15–18 20 21] had used two independent data extractors while one study[14] had not and was classified as high risk of bias for this domain.

## CONSORT harms recommendations

Results extracted for the CONSORT-harms criteria (table 4) demonstrate variability in the level of adherence to items. Heterogeneity is highlighted by the individual Forest plots where inflated $I^2$ values of over 85% are represented for all recommendations, denoting considerable heterogeneity.

Of the six studies that assess inclusion of harms in the title and abstract of their included RCTs, three[16 20 21] reported compliance in over 70% of RCTs, but three[14–16] reported compliance in less than 30% of RCTs. The introduction section of the included RCTs reflect an imbalance in the reporting benefit–harms, with one study[16] reporting that less than 5% of RCTs had mentioned harms in the introduction, and one study[17] reporting more than 70% of its included RCTs has satisfied this criteria.

The definition of adverse events in reports is unsatisfactory with most studies[14–16 17 20] indicating that fewer than 20% of RCTs satisfy these criteria adequately. The collection of harms-related information is described by more than 80% of RCTs in two studies,[20 21] but this high level is not consistent across the other five studies with one study[14] suggesting that as few as 10% of RCTs had provided an adequate description. The analysis and coding of adverse events is poorly described, with less than 50% of RCTs satisfying this criteria across six studies,[13 16–18 20 21] with one of these studies[13] indicating that none of the RCTs had provided an adequate description. The reporting of participant withdrawals due to harms was inconsistent with two studies[15 16] suggesting infrequent reporting (less than 40% of RCTs

**Table 3** Risk of bias assessment

| Risk of bias criteria | Bagul and Kirkham[14] | Breau et al[15] | Turner et al[16] | Shukralla et al[17] | Capili et al[18] | Pitrou et al[20] | Haidich et al[21] |
|---|---|---|---|---|---|---|---|
| Representativeness of sample of trials (low if trials were searched across unselected journals and across a reasonable time period) | High | High | Low | Low | Low | High | High |
| Blinding of reviewers during CONSORT-harms data extraction (low if reviewers blinded to study authors, institution, journal name and sponsors) | High | Low | Low | High | Unclear | High | High |
| Selective outcome reporting (low if all CONSORT-harms criteria assessed) | Low* | Low* | Low*† | Low* | High | High | Low |
| Reliability of data extraction (low if more than one reviewer assessed the CONSORT harms criteria for each review that was undertaken, with a description of how agreement was achieved) | High | Low | Low | Low | Low | Low | Low |

*Recommendation 9 was not included in these studies as subgroup analysis was either not reported in any of the included studies or considered to be irrelevant for the therapeutic area being investigated.
†Authors response: 'Recommendation 8 has been captured elsewhere in data extraction, to report this item would be to duplicate information presented'.
'Recommendation 10 was considered too vague to assess with any objectivity so we decided to leave this item, especially given that some of our primary outcomes were already reasonably subjective'.

**Table 4** CONSORT harms criteria reported across included reviews

| | Bagul (2012)[14] | Breau (2011)[15] | Turner (2011)[16] | Shukralla (2011)[17] | Capili (2009)[18] | Pitrou (2009)[20] | Haidich (2009)[21] |
|---|---|---|---|---|---|---|---|
| Total no. of trials included in review | 41 | 152 | 205 | 152 | 10 | 133 | 102 |
| CONSORT Recommendation | % of trials (95% CI) that adhered to each recommendation | | | | | | |
| (1) Title & Abstract | 20 (9, 35) | 12 (6, 20)<br>1i) 12 (6, 20)<br>1ii) 64 (53, 74) | 21 (16, 27) | 88 (81, 92) | NR | 71 (63, 79) | 76 (67, 84) |
| (2) Introduction | 34 (20, 51) | 54 (43, 65) | 4 (2, 8) | 74 (67, 81) | NR | NR | 48 (38, 58) |
| (3) Definition of adverse events | 0 (0, 9) | 15 (8, 24) | 6 (3, 11) | 3a) 36 (29, 45)<br>3b) 32 (25, 40)<br>3c) 47 (39, 55)<br>3d) 16 (11, 23)<br>3e) 22 (15, 29) | 10 (0, 45) | 16 (10, 23) | 59 (49, 69) |
| (4) Collection of harms data | 10 (3, 23) | 4i) 22 (14, 32)<br>4ii) 6 (2, 13)<br>4iii) 0 (0, 4) | 17 (12, 22) | 4a) 57 (49, 65)<br>4b) 76 (69, 83)<br>4c) 33 (26, 42) | 20 (3, 56) | 89 (82, 94) | 81 (74, 89) |
| (5) Analysis of harms | 0 (0, 9) | 76 (66, 84) | 6 (3, 10) | 5a) 36 (28, 44)<br>5b) 7 (4, 13) | 20 (3, 56) | 12 (7, 19) | 44 (34, 54) |
| (6) Withdrawals | 51 (35, 67) | 35 (25, 45) | 30 (24, 37) | 6a) 71 (63, 78)<br>6b) 72 (65, 79) | 70 (35, 93) | 53 (44, 61) | 59 (50, 69) |
| (7) Number of patients analysed | 17 (7, 32) | 35 (25, 45) | 18 (13, 24) | 7a) 78 (72, 85)<br>7b) 40 (32, 48) | NR | 84 (77, 90) | 74 (64, 82) |
| (8) Results for each adverse event | 39 (24, 56) | 8i) 0 (0, 4)<br>8ii) 28 (19, 38) | – | 8a) 35 (28, 44)<br>8b) 68 (60, 76)<br>8c) 47 (39, 56)<br>8d) 19 (14, 27) | NR | 73 (65, 80) | 89 (82, 95) |
| (9) Subgroup Analysis | – | – | – | – | NR | NR | 53 (43, 63) |
| (10) Balanced discussion | 5 (1, 17) | 10i) 61 (50, 71)<br>10ii) 14 (7, 23)<br>10iii) 44 (33, 55) | – | 10a) 68 (60, 76)<br>10b) 61 (54, 70)<br>10c) 41 (34, 50) | NR | NR | 83 (76, 91) |

NR Not reported in manuscript, and no response from authors when contacted.
– Author detailed reasons for not reporting the recommendation.
1) (i) Harm, safety or similar term used in title; (ii) Harm addressed in abstract.
4) (i) When harm information was collected; (ii) Methods to attribute harm to intervention; (iii) Stopping rules.
8) (i) Effect sizes for harms; (ii) Stratified serious and minor harms.
10) (i) Interpret harm outcome; (ii) discuss generalizability; (iii) discuss current evidence.
3) (a) Definition of AE; (b) All or selected sample; (c) Treatment Emergent AE; (d) Validated instrument; (e) Validated dictionary.
4) (a) Mode of AE collection; (b) Timing of AE; (c) Details of attribution.
5) (a) Details of presentation and analysis; (b) Handling of recurrent AE.
6) (a) Early or late withdrawals; (b) Serious AEs or death.
7) (a) Provide denominators for AEs; (b) Provide definitions used for analysis set.
8) (a) Same analysis set used for efficacy and safety; (b) Results presented separately; (c) Severity and grading of AEs; (d) Provide both number of AEs and number of patients with AEs.
10) (a) Discusses prior AE data; (b) Discussion is balanced; (c) Discusses limitations.

had mentioned withdrawals), three studies[13][20][21] suggesting occasional reporting (50–60% of RCTs had mentioned withdrawals) and two studies[17] suggesting that reporting of withdrawals was quite common (approximately 70% of RCTs had mentioned withdrawals).

When providing the denominators within trial reports, the results were also varied across studies, with three[17][20][21] studies identifying more than 70% of trials that satisfied this criterion, but two studies[13][15] identifying less than 20% adherence. The risk and severity grading of adverse events, is detailed in more than 70% of trial across two studies,[20][21] but the reporting is inadequate in three studies.[13][15][17] An assessment of reporting of harms within subgroup analysis was only carried out within one study.[21]

Four studies[14][15][17][21] assessed their included RCTs for a balanced report on the benefits and harms within their discussion: one study[13] identified a very low percentage (<10%), two studies[14][16] identified a moderate percentage (approximately 60%), and one study[21] identified a high percentage (over 80%) of trials that met this criterion.

## DISCUSSION
### Summary of findings
This is the first study to systematically review empirical studies assessing the quality of reporting according to the CONSORT-harms guideline.[9] Data were extracted from seven studies that had each assessed the quality of reporting across almost 800 RCTs from a range of clinical specialities. Eight years have now passed since the release of the harms extension, allowing adequate time for the guideline implementation. This review highlights that the reporting of harms in RCTs is inconsistent, and at times very poor. Heterogeneity is easily discerned between studies for each recommendation. Further adherence to the CONSORT-harms is needed.

The standard CONSORT is well established in health research with building evidence to support the use of the guideline.[5][6] Currently the standard CONSORT is endorsed by over 50% of the core medical journals in the abridged Index Medicus on PubMed.[22] In a review[23] of 116 health research journals, 41 provided online instructions to authors. Almost half (19/41 (46%)) mentioned the standard CONSORT guideline but none referred to the CONSORT extension for harms.

### Strengths and weaknesses of the study
In this study we have focused on assessing reporting according to the CONSORT-harms criteria only. The included studies contained trials reported prior to the publication of the CONSORT-harms guideline. However, we have not assessed changes in reporting over time. Nevertheless, our results support those from previous studies[3][4] that used various guidelines published before the release of the CONSORT-harms extension. This study should be regarded as a reflection of reporting standards in general rather than an assessment of adherence to the CONSORT-harms extension.

This review was strengthened by its assessment of quality of the included studies across four key domains. With the guidance of the Cochrane review[12] we have designed a RoB tool to perform a generalisable assessment of the included studies. In this assessment only the one study[15] demonstrated low RoB across all four of the assessment criteria. No restriction was placed on the inclusion criteria of the identified studies such that the time span and clinical areas of their included studies varied. While this is a strength in terms of generalisability of results, it may also be considered as a level of heterogeneity that cannot be explored due to the limited number of studies.

### Conclusions and implications
Complete and accurate reporting is essential to guide decisions on advances in medical interventions. The responsibility to ensure greater balance between reporting of both benefits and harms lies with authors of research and journals publishing that research. We recognised that journals have limited space for the reporting of all outcomes which can lead to selective outcomes reporting.[24][25] We recommend the use of supplementary online tables to help summarise key results on harms.

Further dissemination strategies should be used to ensure that trial journal editors and trial investigators are aware of the importance of adequate reporting of harms-related data in RCTs. As it stands, it is unclear as to whether the problem of the poor reporting of harms data in trial publications is a result of the lack of awareness of the CONSORT for harms statement, or journals and peer reviewers not implementing this guideline. The most effective strategy would follow that of the CONSORT statement with the extension for harms comprehensively incorporated in journal requirements along with clear instructions to peer reviewers for guidelines of acceptance.

**Acknowledgements** The authors are grateful to Su Golder and Fiona Beyer York University for expanding on the item words used within the databases and Alison Beamond University of Liverpool, for recommending databases to search conference abstracts.

**Contributors** AH and CTS carried out all screening of literature; AH and JJK extracted data; AH, JJK, CTS and CG interpreted results and drafted the manuscript.

**Funding** This work was supported by the award of a Capacity-Building Studentship to AH from the Medical Research Council (MRC) (grant number G1000397 – 1/1) North West Hub for Trials Methodology Research, UK (grant number G0800792).

**Competing interests** None.

**Patient consent** Obtained.

**Provenance and peer review** Not commissioned; externally peer reviewed.

**Data sharing statement** Data extraction form and protocol available on request from ahoddy@liverpool.ac.uk.

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
