## [Reviewer comments · BMJ Open]

Some articles will have been accepted based in part or entirely on reviews undertaken for other BMJ Group journals. These will be reproduced where possible.

ARTICLE DETAILS

TITLE (PROVISIONAL)	Reporting of harms data in RCTs: a systematic review of empirical assessments against the CONSORT Harms extension.
AUTHORS	Hodkinson, Alex; Kirkham, Jamie; Tudur-Smith, Catrin; Gamble, Carrol

VERSION 1 - REVIEW

REVIEWER	Shannon M. Smith Assistant Professor, Dept of Anesthesiology University of Rochester School of Medicine and Dentistry USA
REVIEW RETURNED	01-Jul-2013

THE STUDY	Inclusion/exclusion criteria: would be helpful to report the cutoff date used for selecting reviews for this project. Methods: it wasn't clear to me why selection of specific journals equates to bias (item 1 under 'quality assessment'), as long as the authors indicate that their results only apply to the selected journals. Reference #12: looks like author Plint AC isn't included
RESULTS & CONCLUSIONS	The discussion focuses only on the findings from the selected reviews, but does not comment on the quality of these reviews, which was assessed and reported in the results. Since the authors took the time to do this important step, it would be valuable to comment on whether the results of the reviews are worth considering (given that they were found to have bias).
GENERAL COMMENTS	This is a valuable effort on the part of the authors. A few minor revisions would help strengthen this manuscript, but otherwise this is an important demonstration of the failure to report AEs in publications of RCTs.

REVIEWER	Yoon K Loke Senior Lecturer in Clinical Pharmacology University of East Anglia
REVIEW RETURNED	03-Jul-2013

THE STUDY	I would like to see further justification for the items listed under Quality Assessment. The current text states that it is an adaptation of the Cochrane ROB tool, but the ROB tool actually designed primarily for between-group comparisons in parallel group RCTs. The present manuscript is a descriptive study of rates of adherence/non-adherence to particular items on a checklist. Hence, there should be some additional text describing why the authors feel that each of their quality items has a bearing on the validity? Some of the items are vague e.g. unselected journals (I'm
------------------	--

	not clear how construction of a cohort of trials in hypertension could be unselected, please give examples), or what is reasonable time scale? Moreover, regarding quality item 3 - selective outcome reporting - I raise the question of whether selective outcome reporting has taken place if the original researchers had defined a priori the outcomes they intended to measure.
RESULTS & CONCLUSIONS	I suggest that Forest Plots (meta-analysis of proportions) could be constructed for each CONSORT Harms recommendation. By graphically depicting the proportions found in each included study, readers can easily discern extent of compliance and heterogeneity between studies. The final pooled proportion is unimportant but the heterogeneity is much more easily discerned in the Forest plot, and the I-squared can be reported.
GENERAL COMMENTS	A couple of other minor suggestions: 1) The conclusion calls for better dissemination of CONSORT Harms, but equally there is an issue regarding implementation. It is not clear at present to what extent the problem stems from lack of knowledge regarding CONSORT Harms, or whether there is actually no will to implement it. 2) The acknowledgements - Fiona Bayer should be spelt as Fiona Beyer.

VERSION 1 – AUTHOR RESPONSE

Reviewer 1

Shannon M. Smith

Assistant Professor, Dept of Anaesthesiology

University of Rochester School of Medicine and Dentistry

USA.

- Comment 1: "Inclusion/exclusion criteria; reporting the cut-off date for selecting the reviews.

Within the identification of studies we have detailed the cut-off date for searching with the primary inclusion of the reviews published after the release of the CONSORT harms document 2004 (P3, L104-106).

- Comment 2: "It was unclear why selection of specific journals equates to bias: item 1 under quality assessment"

High/Low and Unclear risk of bias has now been defined for each quality assessment criteria (Page 4). We judged specific journals to be high risk of bias because the included trials in the studies were highly selective. A primary search of all the available evidence which included both specific and general medical would provide a complete picture of the reporting standards for any particular clinical condition.

- Comment 3: "Plint AC missing from reference 12"

Updated in manuscript (Page 9, Line 333-335)

- Comment 4: "Discussion focuses only on the findings from the selected reviews but does not comment on the quality of these reviews, which was assessed and reported in the results"

Further detail of the quality assessment of the included studies is discussed within the strengths and

weaknesses of the discussion. We highlight that there was only one study that had low risk of bias across all the domains we considered (P8, L260-274).

Reviewer 2

Yoon K. Loke

Senior Lecturer in Clinical Pharmacology

University of East Anglia.

- Comment 1: "Further justification for items listed under the quality assessment. Additional text describing why authors feel that each of their quality items has a bearing on the validity. Some items were vague e.g. unselected journals, or what is reasonable time scale? Question raise about selective outcome reporting item and clarification needed about the interpretation for bias".

In response to reviewer 1 also, the risk of bias assessment items have been detailed in the methods. We have been more transparent about how high/low/unclear risk of bias was assessed for each domain considered (Page 4).

- Comment 2: "Forest plots (meta-analysis of proportions) could be constructed for each CONSORT harms recommendation. Heterogeneity can be discerned in the forest plot, and the I-squared can be reported".

Forest plots have been placed in a supplementary file for readers to access online, with I-squared statistic to describe the amount of heterogeneity; this has also been discussed in the text of the manuscript (P5, L157; P5, L167-174; P6, L210-212; P7, L249-250).

- Comment 3: "Conclusion calls for better dissemination of CONSORT Harms, but equally there is an issue regarding implementation. Not clear at present to what extent the problem stems from lack of knowledge regarding CONSORT harms, or whether there is no will to implement it".
Conclusion section discusses some of the current issues with implementation and highlights possible effective strategies going forward (P8, L277-291).

- Comment 4: "Fiona Bayer should be spelt as Fiona Beyer"
Amendment made in acknowledgements section (P9, L295).